

# How are public compensation efforts implemented in multi-hazard events? Insights from the 2020 Gloria storm in Catalonia

Núria Pantaleoni Reluy[1], Marcel Hürlimann[1], Nieves Lantada[1]

[1]Department of Civil and Environmental Engineering, Universitat Politècnica de Catalunya, Barcelona, 08034, Spain

*Correspondence to*: Núria Pantaleoni Reluy (nuria.pantaleoni@upc.edu)

**Abstract.** Natural disasters result in increasing economic losses worldwide. Existing loss databases primarily capture insured damages and therefore often overlook uninsured assets and public compensation efforts. This study examines the role of public-sector compensation in disaster recovery, using the multi-hazard 2020 Storm Gloria in Catalonia as a case study. By systematically collecting, classifying and analyzing public compensation data related to rebuilding and restoring the direct tangible damages, we provide new insights into financial aid distribution for disaster recovery. In addition, an analysis of 10    single major hazards is performed to understand the event's frequency, as well as its temporal and spatial distribution. Finally, the damages caused by the storm are used to estimate losses based on the probability of the triggering hazard's occurrence. The findings reveal that fluvial and coastal hazards caused over 80% of recorded damages, while meteorological and slope hazards contributed the remainder. Concerning the affected elements, infrastructure sustained the highest losses, followed by economic and social sectors. Rebuilding and reconstruction costs for Storm Gloria were split evenly between 15    fully public and public-private partnerships efforts. Public funding prioritized community assets and critical infrastructure, using hazard-dependent cost assessments and standardized government procedures. Additionally, the study identifies potential multi-hazard municipalities where overlapping hazards intensified damages, highlighting the need for comprehensive disaster documentation. Results also indicate that fully public compensations lack a direct correlation with 20    hazard probability, reflecting prioritization based on recovery needs rather than hazard frequency. The research underscores the critical role of public intervention in disaster risk management and calls for enhanced data standardization to improve loss estimation methodologies in multi-hazard scenarios. Finally, this study contributes to the improve our understanding on disaster loss assessment and provides a framework for future evaluations of government interventions in post-disaster recovery.

## 1 Introduction

Natural disasters have caused significant and increasing losses worldwide. In 2023, the global occurrence of natural disasters and their resulting economic losses exceeded the past 20-year averages, according to the Emergency Events Database, EM-DAT (CRED, 2024). Specifically, 399 natural disasters were globally recorded in 2023, leading to total economic losses of USD 202.7 billion, surpassing the annual average of 369 disasters and USD 196.3 billion in losses for the 2003–2022 period.



This rise is attributed to greater societal exposure and vulnerability, alongside clear evidence of progressively more frequent and intense extreme events driven by global climate change (IPCC, 2022). Adopted in 2015 to address growing concerns about disaster risk and losses, the Sendai Framework for Disaster Risk Reduction emphasizes risk management based on a thorough understanding of risk and the need for accurate economic loss data to mitigate financial impacts of catastrophic events (UNDRR, 2015).

At present, three global and multi-hazard loss databases exist. The Emergency Events Database, managed by the Centre for Research on the Epidemiology of Disasters (Delforge et al., 2023), NatCatSERVICE operated by MunichRe (NatCatSERVICE, 2024) and SIGMA by Swiss Re (Sigma Explorer, 2024). Several authors have compared existing loss databases and found that most primarily capture direct economic losses, focusing on insurance-covered asset sectors (Mazhin

et al., 2021; Moriyama et al., 2018). Relying solely on private insurance data can lead to inconsistent and arbitrary loss assessments across different sectors and an incomplete accounting of total losses (Ladds et al., 2017; Moriyama et al., 2018; Zaidi, 2018).

Nowadays, there is no fully functioning private insurance market for post-disaster recovery worldwide (Kousky, 2019).

When private insurance is either too expensive or unavailable, public-sector involvement becomes necessary. Governments have intervened to ensure the availability and affordability of disaster loss coverage. Since a multitude of costs are not captured in insurance loss databases, Spain's disaster risk management system, primarily public-sector driven, offers a valuable case study for analyzing uninsured asset losses. Government interventions in disaster risk management vary widely around the world. Some are fully public, managed and funded solely by government resources, while others rely on

collaboration, such as public-private partnerships (PPPs) involving direct cooperation between private companies and governmental agencies (Auzzir et al., 2014). In Spain, after catastrophic events, financial resources are allocated through these two channels. The Spanish General State Administration allocates financial resources to various governmental bodies, including ministries, regional departments, and local administrations as public aid (Jefatura del Estado, 2015). Complementing this, the *Consorcio de Compensación de Seguros* (CCS), a public insurance company operating as an

institutionalized PPP, compensates claims with unlimited state financial backing. Henceforth, we refer to "fully public compensations" as those funds allocated by the Spanish General State Administration, and we refer to "PPP compensations" as those provided by the CCS. Spain's complex disaster risk management system, involving multiple actors, ensures a diverse and comprehensive approach to assessing and compensating for losses. However, data on losses remains fragmented due to varying procedures, purposes and documentation standards across institutions, making public data on disaster losses

difficult to gather and standardize (Ballio et al., 2018; Zaidi, 2018). This issue is not unique to Spain but occurs worldwide.

As a result, no official national or regional database consolidates information on fully public funds allocated for disaster loss compensation, although some local or regional databases exist, often developed through government initiatives or research



collaborations. Thus, several studies have developed their own databases using primary sources such as newspapers,
historical documents and technical reports and several hundred national, regional, hazard-based, and sector-specific
databases exist worldwide (Wirtz et al., 2014). For instance, in Spain, regional databases like INUNGAMA, its updated
version FLOODGAMA (Barnolas and Llasat, 2007; Llasat et al., 2014; Llasat-Botija et al., 2024) and Prediflood
(Barriendos et al., 2014), document fluvial floods from historical records, while SMC-Flood covers coastal events (Gil-
Guirado et al., 2019), together spanning up to 50 years of flood data in Catalonia. These studies are highly detailed, focusing
on a single hazard and recording event occurrences, but they lack coverage of multiple hazards and do not provide detailed
fully public financial compensation data for individual assets. Consequently, the CCS loss database offers the most detailed
and accessible loss database, providing asset-based losses at the municipal level.

The CCS loss database has served as a foundation for numerous studies, highlighting the critical role of comprehensive
historical data for depicting natural disaster impacts and hazard trends (Barredo et al., 2012; Cortès et al., 2018b; Llasat et
al., 2014; López-Martínez, 2023), develop risk assessment models (Cortès et al., 2018a; Martínez-Gomariz et al., 2021;
Rivas et al., 2022), validate risk assessment models (Láng-Ritter et al., 2022; Ritter et al., 2020, 2021; Romero-Martín et al.,
2024), estimate susceptibility curves (Martínez-Gomariz et al., 2020; Mediero et al., 2021) and evaluate losses projections
(Cortès et al., 2019; Martínez-Gomariz et al., 2019; Ribas et al., 2020; Soriano et al., 2023). However, there is a lack of
evidence that historical compensation loss data from other forms of government intervention have been similarly examined,
particularly for fully public compensations, both in Spain and globally. Evaluating this type compensation could therefore
provide valuable insights into these gaps, especially in addressing uninsured sectors, such as environmental losses, essential
facilities like water and electricity systems, and non-essential facilities including cultural and recreational assets, and more
(Ladds et al., 2017).


As a major multi-hazard, high-impact event, the Gloria storm, which struck Catalonia from January 19 to 24, 2020, activated
all public sector post-disaster compensation programs, making it a perfect case study for evaluating the role of public
interventions in disaster response. Marked by persistent and intense rainfall along with strong winds (Martín-Vide, 2020),
the storm event led to a notable increase in sea levels intensified by large waves (de Alfonso et al., 2021; Amores et al.,
2020; Pérez-Gómez et al., 2021), triggering numerous slope failures (Palau et al., 2022), and causing extensive pluvial and
fluvial floods (González et al., 2020; Martín-Vide, 2020). The aftermath of these events resulted in widespread substantial
direct economic losses (Amores et al., 2020). While various studies have examined the Gloria storm, only Sancho-García et
al. (2021) evaluates regional-scale losses by analyzing news reports, combined with hydrodynamic data he identifies coastal
damage hotspots.


As shown by the previous literature review, comprehensive loss databases provide valuable insights into risk understanding.
Enhancing disaster loss databases is a priority at both international and European levels. However, reliance on insurance data



often overlooks uninsured assets, leading to significant knowledge gaps. While public sector compensation data could offer a broader perspective, its collection, organization, and analysis present significant challenges and remain largely unexplored. Therefore, the present study aims to apply an integrated approach to better understand: (i) the role of government interventions in disaster loss response, (ii) insights into multi-hazard risks derived from a loss database, and (iii) the assessment of losses as a function of annual occurrence probability. The study focuses on public loss compensation for uninsured assets across multiple hazards at a regional scale, using the 2020 Gloria storm as a case study. In addition, a comprehensive hazard episode analysis has been performed.

## 2 Case study area and Gloria Storm episode

Catalonia, with its large and expanding urban settlements, dense critical infrastructure, and significant economic activities, is highly vulnerable and exposed to natural hazards due to its spread across a diverse geography, including coastal zones, river valleys, and mountain areas. The geographical and temporal extent of Storm Gloria, combined with its multi-hazard nature, caused severe impacts, making it an ideal case study for examining the challenges and consequences of multi-hazard events in exposed regions.

Catalonia, a region covering approximately 32,000 km², is situated in the northeastern corner of the Iberian Peninsula. Its northern boundary is defined by the Pyrenees Mountain range, with peaks exceeding 3000 m a.s.l. The region's main topography features include a littoral mountain range, with elevations under 500 m a.s.l., running parallel to the Mediterranean coast, and a pre-littoral range further inland, rising between 1,000 and 2,000 m a.s.l. Catalonia's rainfall patterns are influenced by its diverse orography and the proximity to the Mediterranean Sea. Coastal areas experience intense precipitation during spring and autumn months, with annual rainfall ranging from 500 mm in the south to over 900 mm in the north (Mira et al., 2017; Turco and Llasat, 2011). In contrast, inland regions, sheltered from maritime influences, receive significantly less precipitation, with some areas recording below 400 mm annually, the lowest in Catalonia. The Pyrenean region, however, experiences the highest annual rainfall, exceeding 1000 mm, primarily during the summer months (Turco and Llasat, 2011). Focusing on the locally sourced rivers, the network includes major regulated rivers, such as the *Ter* and *Llobregat*, and non-regulated rivers like the *Tordera*, *Fluvià*, *Besòs*, and *Muga* (Fig. 1), all draining to the eastern coast. Bordered by the Mediterranean Sea to the east, the Catalan coastline, which stretches about 600 km long, exhibits diverse landscapes, varying from rocky shores with high cliffs in the north to expansive flat areas with long sandy beaches in the south.



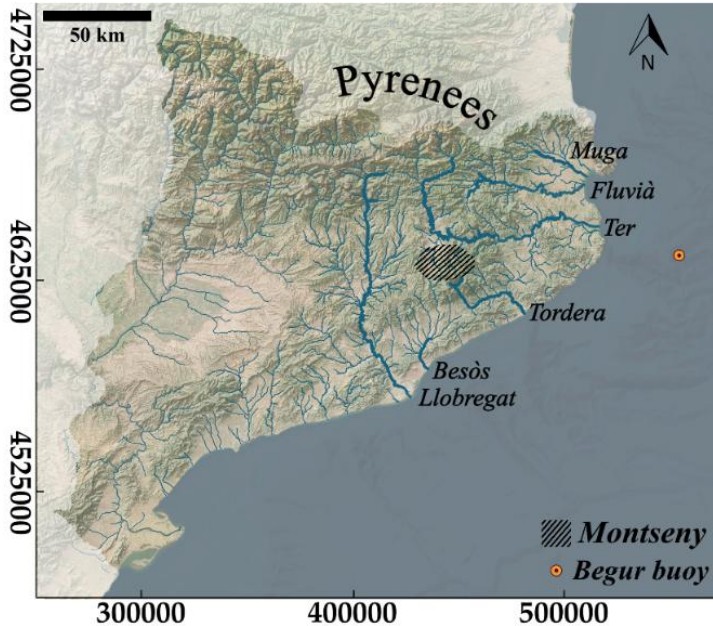

**Figure 1. Geographic overview of Catalonia, indicating main hydrological and topographical features mentioned in the text. Coordinate system: ETRS89 UTM Zone 31N.**

On January 18, 2020, a North Atlantic cold front entered through the northwestern region of the Iberian Peninsula, advancing southward towards the Mediterranean Sea. Simultaneously, an unusual anticyclonic system over the British Isles produced pressures of 1050 hPa, the highest since 1957 (Palau et al., 2022; SMC, 2021). This high-pressure system extended across a substantial portion of central Europe. The formation of the Gloria storm resulted from the pressure gradient between the unprecedented high pressures over the British Isles and the low pressures situated south of the Iberian Peninsula.

Occurring from January 19th to 24th, 2020, Storm Gloria brought persistent and intense rainfall, along with strong winds. This resulted in widespread and significant fluvial floods, a significant sea-level rise enhanced by large waves, and multiple slope failures. To visualize the diverse hazards associated with the storm, Fig. 2a illustrates the hourly precipitation histogram and daily rainfall accumulation, with a remarkable 400 mm of accumulation recorded in the *Montseny* area (Corral et al., 2008). The water discharge in the regulated *Ter* river experienced maximum local discharges of 700 cubic meters per

second (Fig. 2b, ACA, 2024). The *Begur* buoy in the northern coastal zone registered maximum wave heights of nearly 14 meters (Fig. 2c Gómez Lahoz and Carretero Albiach, 2005). Additionally, the daily number and event accumulation of landslides in Catalonia during the Gloria storm reached approximately 270 landslides regionally (Fig. 2d, Palau et al., 2022).





**Figure 2. Temporal evolution of individual hazards during Gloria Storm. a) Hourly and cumulated precipitation in *Montseny***
**145 area; b) *Ter* River water flow; c) Maximum and significant wave height at *Begur* buoy and d) Daily and cumulated landslide**
**occurrence. See Fig. 1 for locations.**

## 3 Methods and data

This study conducted a regional disaster loss analysis by evaluating public sector interventions using a three-step

methodology. First, we compiled a comprehensive loss database, categorizing the public funds allocated for recovery after

the Gloria storm in Catalonia. Second, we analyzed and assessed the single major hazards observed during the Gloria storm

event. Finally, we examined the interfered damages caused by the storm conditions to illustrate the impact assessment of

uninsured assets.



**3.1 Direct loss analysis**

The data collection process involved gathering publicly accessible information on public-sector funds allocated for post-
disaster recovery. The study focused on direct tangible costs, which are defined by Meyer et al. (2013) as financial losses
resulting from direct physical contact with a hazard, quantifiable in monetary terms and closely tied to the specific time and
location of the event. Specifically, we examined data related to the repair and rebuilding of damages, considering the asset's
depreciated value at the time of damage, excluding any improvements (Merz et al., 2010). Henceforth, the terms "costs" or
"losses" will specifically denote the direct tangible costs related to the repair and rebuilding of damages.

The collected data was classified by adapting the Joint Research Centre's (JRC) loss database model methodology (De
Groeve et al., 2013). The methodology was modified based on the goal of assessing losses at the asset level. The structure of
the loss database includes five attributes, each classified into distinct categories. Each entry in the database corresponds to
the direct tangible cost of repairing and rebuilding a specific asset, compensated by public funds, and is described according
to these five attributes. In detail, the attributes considered consist of:

- Geographical location: Denotes the location of the damaged asset. This parameter is defined using the European
  Union's Nomenclature of territorial units for statistics (NUTS) reference classification system. For asset-level
  analysis, we employed the lower level of Local Administrative Units (LAU2) comprising municipalities or
  equivalent units within the EU Member States (Eurostat, 2024).
- Affected element: Defines the asset damaged as a result of the hazard event. The classification system employed is
  adapted from the Damage and Loss Assessment (DaLA) methodology, developed by the Economic Commission for
  Latin America and the Caribbean (ECLAC). DaLA provides a sectoral framework for assessing the social,
  economic, and infrastructural impacts of natural disasters, with each further subdivided (ECLAC, 2003). The
  classification scheme for the "Affected element" attribute is presented in Table 1.
- Hazard identification: Identifies the specific event that triggered the loss. The UN Office for Disaster Risk
  Reduction and the International Science Council established a common hazard classification scheme and definitions
  (Murray et al., 2020). The resulting classification includes 302 hazard information profiles (HIPs), grouped into
  eight hazard types, each further divided by cluster type and by specific hazards. Based on the HIPs classification of
  specific hazard and supplemented by insights from a thorough review of the literature on the main hazards affecting
Catalonia (see Sect. 2), we employed four distinct categories: Fluvial, Coastal, Landslide, and Meteorological
  categories correspond to the "Fluvial (Riverine) flood", "Coastal Flood", "Rock Slide", and "Flash Flood"
  definitions in the HIPs, respectively.
- Cost: Represents the monetary value required for repairing or rebuilding the incurred damage.



- Source: Identifies the institution, or company providing the information from which the damage loss data originates and thus, the entity who bears the loss. We divided this attribute into two categories: "Public-private partnership" and "Fully public".

To maintain a simplified and standardized use of the framework, four attributes are described as qualitative units represented by classes based on adapted existing classification methods ("Geographical location", "Affected element", "Hazard identification" and "Source"), while the "Cost" attribute is represented by a numerical value.

**Table 1. Classification scheme for characterizing the attribute "Affected element" (adapted from ECLAC 2003).**

| First-level categories | Second-level categories | Description |
|---|---|---|
| Social | Residential | Buildings used as dwellings and their contents.<br>*E.g., Housing, domestic furniture, equipment…* |
| | Education / research | School or educational premises and auxiliary installations.<br>*E.g., Teaching premises, administrative buildings…* |
| | Culture and recreation | Sites considered as cultural and historical heritage.<br>*E.g., Museums, churches, public areas* |
| | Health | Infrastructure of the health-care services network.<br>*E.g., Hospitals, laboratories, blood banks…* |
| Infrastructure | Energy | Electrical and oil infrastructures.<br>*E.g., Generation plants, distribution network…* |
| | Water | Components of water and sanitation systems.<br>*E.g., Drinking water supply, sanitation, sewerage…* |
| | Transport | Transportation network.<br>*E.g., Road, harbor, airport…* |
| | Communication | Telecommunications services.<br>*E.g., Telephony, radio and television broadcasting…* |
| Economic | Agribusiness | Farm infrastructure and operational assets.<br>*E.g., farmland, equipment, production losses…* |
| | Industry | Establishments where industrial activities are conducted.<br>*E.g., Factories, warehouses, machinery…* |
| | Trade | Businesses involved in commercial activities.<br>*E.g., storefront, furniture, stocks or inventories…* |
| | Tourism | Premises for tourism activities and related services.<br>*E.g., Hotels, tourism facilities…* |
| Environmental | Environment | Environmental ecosystems goods and service<br>*E.g., Forest, coastal, freshwater ecosystems* |
| Other | Vehicle | |
| | Office | |

**3.2 Hazard analysis**

A multifaceted approach is employed to provide a comprehensive understanding of the hazard episode, focusing on three aspects: frequency, temporal and spatial extent. Frequency is evaluated by calculating return periods to assess the severity of





the event compared to historical occurrences, while temporal duration is determined by quantifying how long the storm persisted. These aspects are then integrated with a spatial analysis. This procedure was applied exclusively to meteorological, fluvial, and coastal hazards. Due to the inherent complexity of landslides mechanisms, the lack of reliable historical data, and the need for specific site-based assessments, this procedure was not applied to landslide hazard.

Return period is a statistical estimate of how often an event exceeding a certain intensity is likely to occur, based on historical data. Extreme Value Theory (EVT), introduced by Fisher and Tippett (1928), is commonly used to analyze the probability of rare events. The Generalized Extreme Value (GEV) distribution, developed within EVT, approximates the probability distribution of long sequences of independent extreme variables (Coles, 2001). The three-parameter GEV distribution combines Gumbel, Weibull, and Fréchet distributions into a single expression. The cumulative distribution

function (CDF) of the GEV is described according to Jenkinson (1955) as:

$$F(z) = \begin{cases} \exp\left\{-\left[1 + \xi\left(\dfrac{z-\mu}{\psi}\right)\right]^{-1/\xi}\right\} & \xi \neq 0 \qquad \text{Eq. (1a)} \\[2mm] \exp\left\{-\exp\left[-\left(\dfrac{z-\mu}{\psi}\right)\right]\right\} & \xi = 0 \qquad \text{Eq. (1b)} \end{cases}$$

Where, $\mu$ is the location, $\psi$ is the scale and $\xi$ is the shape parameter. Defining the return period as the inverse of the exceedance probability, for a specific return level $z$, the return period ($T_z$) is given by $T_z = \dfrac{1}{(1-F(z))}$, which results in (Palutikof et al., 1999) :

$$z = \mu - \psi \ln\left\{-\ln\left(1 - \dfrac{1}{T_z}\right)\right\} \quad \xi \neq 0 \qquad \qquad \text{Eq. (2)}$$


Based on the EVT and the precedents equations, return periods for major hazards can be calculated. In this study, we apply this methodology to characterize the Gloria storm, considering three main hazards: meteorological, fluvial, and coastal. The following sections describe the method applied to each of these hazards.

### 3.2.1 Meteorological Hazard

Meteorological data encompasses (1) rainfall data and (2) Intensity-duration-frequency (IDF) rainfall data (Table 2). Precipitation estimates were derived from the rain gauge and weather radar network observations of the Meteorological Service of Catalonia. These observations were integrated using the method proposed by Velasco-Forero et al. (2009) and Cassiraga et al. (2020), which employs the geostatistical technique known as kriging with an external drift to interpolate rain gauge data with radar rainfall serving as the secondary variable. The hourly rainfall data covers the period from 19th January

to 23rd January 2020, spanning the entire Catalan region with a spatial resolution of 1 km. This data was provided by Center of Applied Research in Hydrometeorology (CRAHI). In addition, IDF rainfall data for selected durations and return periods




were obtained from the study conducted by Llabrés Brustenga (2020). The dataset consists of 40 raster maps with a 1 km spatial resolution, displaying expected rainfall accumulation for various durations (1h, 6h, 12h, 24h, and 48h) and for different return periods (2, 5, 10, 20, 50, 100, 200, and 500 years). These maps are provided by the Catalan meteorological service and are freely available on their website (Meteocat, 2023).


The expected rainfall intensity maps were derived using a Gumbel distribution fitting (Llabrés Brustenga, 2020). Therefore, according to Eq. (2), the rainfall accumulation ($z$) for the different return periods and $-\ln\left\{-\ln\left(1 - \frac{1}{T}\right)\right\}$ follow a linear distribution when considering a specific rainfall accumulation time interval (Fig. 3). To determine the return period of the

rainfall during the Gloria event in each pixel, we compared the observed maximum rainfall of the event with the linear distribution. The return period was assigned based on the return period exceeded by each 1 km resolution pixel (2, 5, 10, 20, 50, 100, 200, and 500 years). For instance, with 123 mm of rainfall accumulation over 48 hours during the Gloria event, the estimated return period is 29 years, but it is assigned as 20 years as it surpasses the 20-year threshold (Fig. 3).

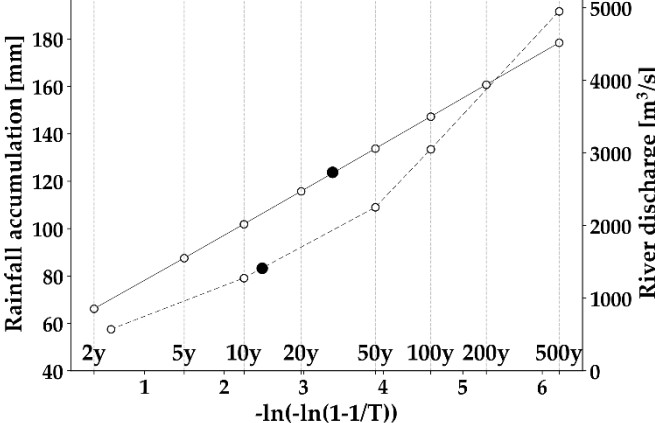


**Figure 3. Example of return period estimates for meteorological (continuous line) and fluvial hazards (dashed line). The filled dots correspond to the Gloria storm event. 48-hour rainfall accumulation was selected for the meteorological hazard, and the observed discharge water level at the *Llobregat* river for the fluvial hazard.**

### 3.2.2 Fluvial Hazard

Hydrological data includes (1) in-situ discharge measurements and (2) discharge reference levels corresponding to return period values (Table 2). The data cover 53 gauging stations distributed across 6 internal basins in Catalonia (from north to south: *Muga*, *Fluvià*, *Tordera*, *Ter*, *Besòs* and *Llobregat* river basins, see Fig. 1). Only rivers originating and flowing within Catalan watersheds are monitored by Catalan Water Agency (ACA) stations, while those beyond these boundaries fall under the *Confederación Hidrográfica del Ebro*'s management. For consistency and due to a lack of discharge reference data for

these external rivers, they were excluded from this analysis. The distribution of gauging stations is as follows: 19 along the *Llobregat*, 16 in the *Ter*, 4 in the *Tordera*, 7 in the *Muga*, 4 in the *Fluvià*, and 3 in the *Besòs*. Of these, 39 stations are



situated on non-regulated river branches, with the remaining 14 on regulated ones. In-situ discharge records, collected between January 20th to 26th, 2020, were recorded at varying frequencies: 40 stations provided 5-minute measurements, while 13 collected 15-minute data. Stations with missing data due to sensor damage caused by the Gloria storm were
excluded. For discharge reference levels, data were acquired corresponding to 5 reference return period values (2.33, 10, 50, 100 and 500 years). All data were provided by the ACA and are freely available upon request.

Based on the assumption of a Gumbel distribution, different linear regressions were fitted for each reference discharge level ($Q$) and $-\ln\left\{-\ln\left(1-\frac{1}{T}\right)\right\}$ for each specific gauge station (Fig. 3). To assess the return period of fluvial water discharge
during the Gloria event, observed maximum discharge data is compared with the linear regressions. For instance, with a maximum Gloria discharge water level of 1411 m$^3$/s observed over the *Llobregat* river, the return period estimated from the linear regression is 12 years (Fig. 3).

### 3.2.3 Coastal Hazard

The significant wave height data employed in this study consist of numerical modeling data from the SIMAR-WANA
database (Gómez Lahoz and Carretero Albiach, 2005), provided by the Spanish port authority (*Puertos del Estado*). This database merges subsets from two different computational models: SIMAR-44 and WANA. Atmospheric and wave conditions in the Mediterranean were simulated using the WAM model for the SIMAR-44 database, providing wave data from 1958 to 2000 (Pilar et al., 2008). The database was later extended with the WANA database, providing wave data from 2000 onward. The open-source data repository offers hourly significant wave height data, covering the period from January
1, 1958, to the present freely accessible online (Puertos del Estado, 2024). For the purposes of this research, significant wave height data from 52 grid nodes along the coastline of Catalonia were considered (Table 2). Among these sites, 29 are located within 5 km of the Catalan coastline and 23 are situated approximately 10 km from the coast.

To calculate the return period of the coastal hazard based on this comprehensive significant wave height dataset, the classical
Block Maxima (BM) approach was employed to extract extreme values by partitioning time series into equal-duration blocks (Coles et al., 1999). In this study, annual maximum significant wave heights were extracted for the period spanning from 1958 to 2023, resulting in a dataset comprising 65 observations. These extreme values, assumed to be identically distributed and stochastically independent across blocks, were fitted using the GEV model. Parameter estimation for the GEV model was performed using the maximum likelihood method, where the likelihood function is maximized to determine the
parameters that best fit the observed data (Myung, 2003). For the Gloria event, return periods of simulated maximum significant wave height at each SIMAR grid node were obtained by inverting the fitted GEV cumulative distribution function (Eq. (2)). The convergence quality of the GEV distribution was evaluated using the Kolmogorov-Smirnov test (Lucio, 2004), with a significance level of 5 %.



**Table 2. Selected hazards and information on dataset**

| Hazard | Parameter | Source | Type | Spatial and temporal resolution | Data time period |
|---|---|---|---|---|---|
| Meteorological | Rainfall | CRAHI | Estimation | 1 km Hourly | From 19 to 23 January 2020 |
| | IDF | Llabrés Brustenga (2020) | Estimation | 1 km | Return period 2, 5, 10, 20, 50, 100, 200 years for rainfall accumulation of 1h, 6h, 12h, 24h, 48h |
| Fluvial | Discharge | ACA | In-situ | 53 stations 5 or 15 minutes | From 20 to 25 January 2020 |
| | Discharge reference levels for return periods | ACA | Estimation | 53 stations | Return period 2.33, 10, 50, 100 and 500 years |
| Coastal | Significant wave height | *Puertos del Estado* | Estimation | 52 grid nodes Hourly | From 1 January 1958 to 1 January 2023 |

## 4 Results

### 4.1 Direct damages analysis

#### 4.1.1 Loss compensation distribution

This section presents how public compensation was distributed to cover a multi-hazard event focusing on the economic

impacts of the Gloria storm in Catalonia. Using a comprehensive database categorized into five distinct attributes, we present on one side the density distribution of all allocated compensations (Fig. 4), and on the other side the general distribution and interconnections among attributes, highlighting the entities involved in managing hazard-related event recovery (Fig. 5).

The Gloria storm caused substantial financial impacts, with total repair and rebuilding costs recorded at EUR 264,301,110,

based on the compiled database of public loss compensations. As shown in Fig. 4, the median damage cost for the entries in the database was approximately EUR 14,000, with the highest compensation exceeding EUR 10 million, and the lowest recorded as EUR 28. Fully public funds typically covered moderate costs, with a median compensation nearly doubling that of PPPs, which spanned a wider range including both larger and smaller amounts. "Coastal" hazards incurred the highest median cost (EUR 30,000), with a wide distribution slightly skewed towards higher losses, and the upper quartile reaching

up to EUR 100,000. In contrast, "Fluvial" hazards, despite being responsible for a larger share of the damage, showed costs clustering around a median of EUR 20,000, suggesting numerous smaller-scale damages or distinct compensation criteria. "Meteorological" hazard damages received the lowest median compensation (around EUR 10,000), possibly due to lower overall damages.



Both "Environmental" and "Infrastructure" first-level "affected element" categories received higher median loss compensation amounts (EUR 20,000). Additionally, the highest compensation costs were observed for "Economic" and "Infrastructure", where the upper quartiles reached EUR 45,000. Notably, "Environmental" showed a narrower distribution, with a EUR 6,000 difference between the upper and lower quartiles, and by contrast "Economic" had a broader distribution. This variability in compensation was largely driven by the different types of second-level "affected element" in each first-

level. For example, in "Infrastructure", high costs were mainly attributed to the "Transport" element, while in "Economic", compensation levels varied between affected element categories like "Agribusiness", which incurred lower costs, and "Industry" and "Trade", which were costlier. Additionally, the cost compensation variability could also have stemmed from different compensation criteria, as seen and suggested by the narrow distribution for the "Environmental" element.

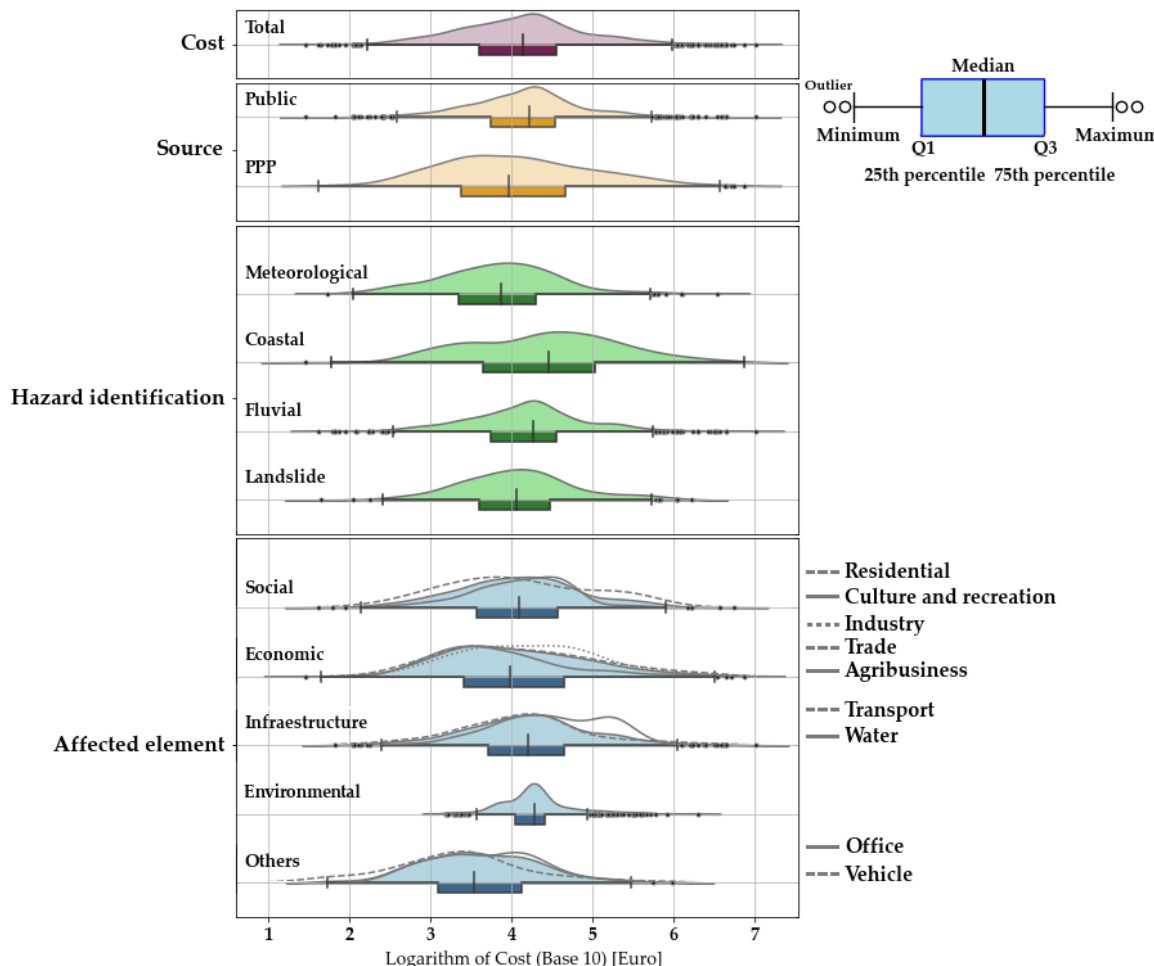

**Figure 4. Density distributions of public compensation allocations for direct losses across the attributes "Cost", "Source", "Hazard identification" and "Affected element" for the Gloria storm. Upper part is shown as a density plot and the lower part as a boxplot. Regarding "Affected element": The filled areas correspond to the first-level categories (see Table 1) and the unfilled areas represent second-level categories (see legends at the right), represented solely by the upper density plot.**



The following points describe the findings regarding the distribution of damage, funding sources, and sector-specific recovery efforts after the Gloria storm:

- The primary damage triggers were "Fluvial" (43%) and "Coastal" (39%) hazards, which together accounted for over 70% of total losses across all examined "Affected element" firs-level categories. Additionally, "Meteorological" and "Landslide" hazards each contributed 9% of total losses. This difference further reflects the lower overall cost of the damages as well as a lower prioritization of "Meteorological" and "Landslides" hazards. Nonetheless, the findings highlight Catalonia's significant exposure and vulnerability to coastal and fluvial hazards. For most hazard types, rebuilding and reconstruction costs were evenly split between the two funding sources, reflecting a balanced sharing of financial responsibilities. However, landslide-related damages were almost entirely funded by fully public sources, underscoring the difficulty of involving PPP in addressing these specific events.

- By affected elements, "Infrastructure" sustained the largest losses (42%), followed by the "Economic" (28%) and "Social" (20%), with "Environmental" and "Other" accounting for the remaining 10%. The allocation of funding from fully public and PPP sources across affected elements highlighted the targeted nature of the recovery process. Fully public funds predominantly supported "Infrastructure" and "Environmental" elements, while PPP funding was more prevalent for "Economic", "Social", and "Other". Fully public funds median value aligns with the distribution of costs in the "Infrastructure" and "Environmental" categories (Fig. 4), whereas PPP funding covered a wider range of costs, matching the distribution of losses in the "Economic", "Social", and "Other" elements.

- A deeper analysis of second-level categories highlighted how compensations were distributed among "Affected elements". Fully public funding played an essential role in addressing broader community assets and essential services, underlining the reliance on public resources for recovery in sectors directly affecting public welfare and vital infrastructure. These funds were entirely allocated toward sectors such as "Agribusiness", "Environment", "Water", "Culture and Recreation" and "Transport". Conversely, PPP funding targeted sectors like "Trade", "Industry", "Vehicles", "Residential" and "Offices", which may have been more manageable for public-private involvement. It is important to note that no data was recorded for "Energy", "Education/Research", "Health", "Communication" and "Tourism".





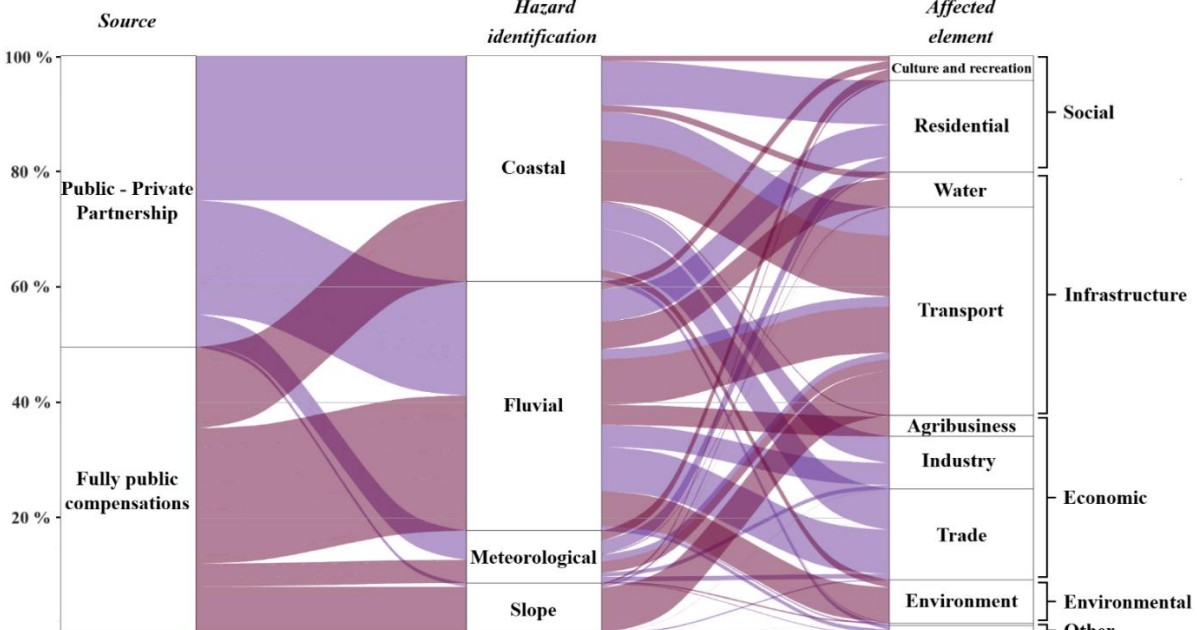

**Figure 5. Relation between direct damage costs by "Source" (left column), "Hazard identification" (middle column) and "Affected element" (right column) triggered by Gloria Storm.**

Gloria storm recovery was addressed through a nearly even combination of fully public and public-private partnership funds (Fig. 5). The findings reflected a strategic response to recovery, showing that while funding sources were not directly linked to hazard type, they were closely tied to the "Affected element". Moreover, the analysis of fully public funds revealed that hazard impacts were sector-specific, with certain hazards affecting specific elements exclusively. For instance, all recorded losses in "Agribusiness", "Environment", and "Water" were attributed to "Fluvial" hazards. Similarly, "Slope" hazards primarily affected the "Infrastructure" sector, while "Meteorological" hazards were strongly associated with the "Social" sector, particularly in "Culture and Recreation", where heavy rainfall disproportionately damaged public and recreational assets. This pattern, however, was not visible in PPP funding. These findings suggested that different compensation criteria were applied depending on the funding source. Fully public funds prioritized hazard type first, followed by the affected element, whereas PPP funding focused solely on the affected element, regardless of the hazard type.

### 4.1.2 Geographical distribution

This section analyses the geographical distribution of the public-sector compensations. Regarding the total compensation data collected, 94% of the entries include information at the municipal level, covering 70% of Catalan municipalities (total of 947) (Fig. 6a). Among these, 37% incurred losses under EUR 25,000, predominantly in inland areas, 29% faced compensations between EUR 25,000 and EUR 100,000, and 34% experienced damages exceeding EUR 100,000, mostly in coastal regions. It's important to note that 8% of municipalities exceeded EUR 1 million damages.





**Figure 6. Geographical distribution of the loss compensation data at municipality scale for the Gloria storm. a) Total losses; b) Major compensation source; c) Most damaging hazard type; d) Number of damaging hazards; e) Most damaged "Affected element"; f) Number of "Affected element".**



Public-Private Partnership funds were the main source of aid in 38% of municipalities, with 36% relying exclusively on them (Fig. 6b). In contrast, fully public funds were the predominant form of aid in 62% of municipalities, with over half (53%) depending solely on this aid. This suggests that while PPPs contribute to recovery, fully public funding is the primary support in more municipalities. Generally, PPP compensation focused along coastal and northern inland areas, while public funds were focused on inland regions, likely due to greater public responsibility or limited PPP sector involvement in these areas.

The hazard-related damage analysis shows that fluvial damages were the most widespread, being the primary hazard in 50% of municipalities, particularly in the northeast area of Catalonia near the *Muga, Ter*, and *Tordera* rivers (Fig. 6c). Slope-related hazards, concentrated near the Pyrenees, impacted majorly in 14% of municipalities, while meteorological damages affected 26%. Coastal effects were predominant in nearly all coastal municipalities, accounting for 10% of the municipalities. Regarding the spatial distribution of the "Affected element", compensation for "Transport", "Residential", and "Culture and Recreation" damages were the most extensive, accounting for over half of the total damage costs in 27%, 21%, and 18% of municipalities, respectively (Fig. 6e). The municipal-scale map accurately depicts significant impacts on "Residential" and "Culture and Recreation" in major cities, "Industrial" near urban centers, and widespread damage to "Transport" across the region, underscoring the critical importance and exposure of these infrastructures.

We define potential multi-hazard municipalities as areas where losses resulting from more than one hazard are recorded. The analysis reveals that 66% of municipalities experienced losses from a single hazard, 26% from two, and 8% from three (Fig. 6d). Despite these municipalities showing a form of multi-hazard occurrence, in 96% of municipalities a single hazard was responsible for the majority (over 70%) of compensated losses. This suggests that although multiple hazards may affect a region, one event tends to dominate in terms of damage. The distribution of damages across different affected elements further underscores this. Specifically, 32% of municipalities reported losses from a single affected element, and 23% experienced losses from two. However, aligning with the geographical distribution of the potential multi-hazard municipalities, 45% recorded damages across three to nine assets (Fig. 6f). Additionally, in 60% of municipalities, a single asset accounted for more than half of the compensated damage costs. Notice that, there is a consistent geographical relationship between municipalities affected by multiple hazards and those with a high number of impacted elements, leading to more severe overall damage. Primarily concentrated near river northeastern areas, Barcelona city and the deltaic zone of the Ebro River in the south. However, this geographical correlation did not appear to influence the sources of compensation funding.



## 4.2 Hazards analysis

### 4.2.1 Meteorological Hazard

The spatial distribution of maximum rainfall accumulation over 1, 12, and 48-hour durations during Gloria Storm and the corresponding return periods was calculated as described in Sect. 3.2. and is shown in Fig. 7. While the peak rainfall
intensity was not exceptional (reaching 75 mm in 1 hour), the storm's extended duration led to significant maximum accumulations (181 mm in 12 hours and 398 mm in 48 hours). Over the Catalan region, mean rainfall accumulation increased from 21 mm for 1-hour accumulation, to 63 mm for 12-hour and 127 mm for 48-hour accumulations. Longer rainfall accumulation periods exhibited higher return periods, suggesting that although the rainfall intensity was not extreme compared to historical records, the extended duration made the event uncommon. Specifically, within 1 hour, 99% of the
Catalan region recorded accumulations below 50 mm, consistent with historical data resulting in return periods of less than 2 years for most of the region (Fig. 7d). Over 12-hour accumulation period, approximately 40% of the region recorded less than 50 mm, with these areas not exceeding the 2-year return period (Fig. 7e). However, the remaining areas, which experienced rainfall accumulations surpassing 150 mm (13%), had return periods ranging from 5 to 50 years, notably in the northeastern, southwestern, and central regions (Fig. 7e). Similarly, for the 48-hour accumulation period, 73% of the territory
observed rainfall accumulation below 150 mm, corresponding to return periods of 2 to 5 years (Fig. 7f). However, the remaining areas (31% of Catalonia) reached diverse return periods, ranging from 10 to more than 500 years, with rainfall accumulations ranging from 200 to nearly 400 mm, particularly in the northeastern, southwestern, and central areas (Fig. 7f).








**Figure 7. Geographical distribution of maximum rainfall accumulation (top) and return periods (bottom) for a duration of 1 hour (left), 12 hours (middle) and 48 hours (right) related to the meteorological hazard analysed.**

The storm brought remarkably substantial precipitation across Catalonia regarding the entire episode, marking it as an exceptionally rare occurrence for the month of January. When considering individual affected areas, there have been relatively recent episodes with comparable or higher rainfall accumulated quantities, but the widespread nature of this event hasn't been observed since the first half of the 20th century (Canals and Miranda, 2020). Catalonia experiences approximately six episodes of extreme torrential precipitation per year, each accumulating 100 mm or more in 24 hours, predominantly during autumn (López-Bustins and Martín-Vide, 2020). Historical records document few instances of continuous torrential rain in Catalonia in January, specifically from January 15 to 18, 1898 (Barriendos et al., 2014).



Therefore, the occurrence of a storm event like Gloria in the first month of the year has a return period exceeding a century. As for daily accumulation, only two January episodes during 1951 and 2016 exceeded 200 mm, equivalent to the intensity of the Gloria storm (January 6, 1977 and January 29, 1996) (Lopez-Bustins et al., 2020).

### 4.2.2 Fluvial hazard

The spatial distribution of the fluvial discharge return period reached during Storm Gloria at 53 monitored gauge stations along the Catalan internal watersheds is shown in Fig. 8. Additionally, the figure illustrates the accumulated number of hours exceeding the return periods of 10, 15 and 20 years during the storm event along *Llobrega*t and *Ter* rivers.

Out of the 53 monitored gauge stations, 20 remained below the 5-year discharge return level. These stations were primarily
situated in unregulated upstream, and tributary sections of the regulated rivers *Llobregat, Ter*, and *Muga*. In contrast, along the main channels of these regulated rivers, which are controlled by reservoirs, discharge records showed return periods between 10 and 20 years, with only a few instances exceeding the 20-year return period in the *Ter* river. Reservoir management played a critical role during the storm, significantly influencing discharge levels and the duration of exceedances (Fig. 8). In the *Llobregat* river, this management led to an average of 40 hours of exceedance for smaller return
periods (10 years), while in the *Ter* river, reservoir management resulted in longer exceedance durations (60 hours) and higher return periods downstream. Conversely, nearly all stations on the unregulated rivers, such as *Fluvià, Besòs*, and *Tordera*, exceeded the 5-year discharge level. The unregulated sections of the *Tordera* and *Fluvià* rivers recorded the highest number of stations surpassing the 20-year discharge return level.




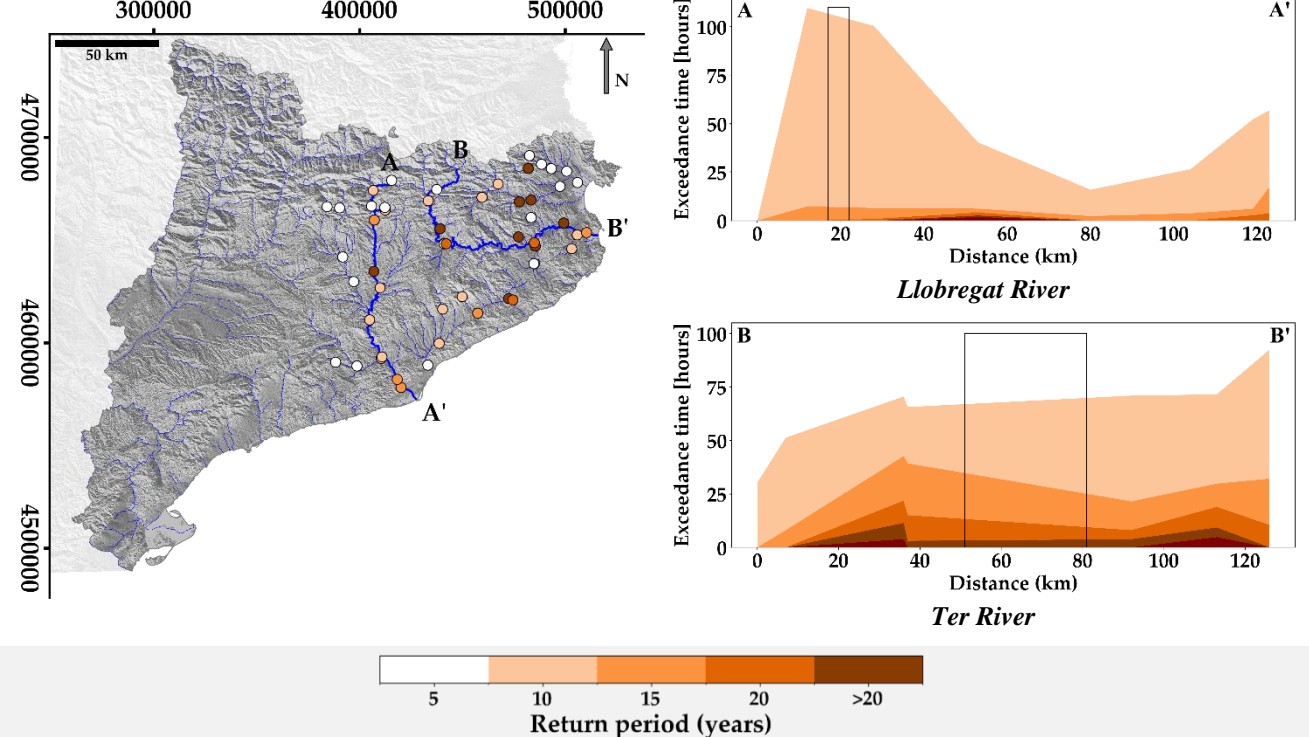

**Figure 8. Geographical Distribution of the water discharge return periods related to the fluvial hazard analysed and the hours of exceedance for the *Llobregat* (A-A') and *Ter* (B- B') Rivers. Vertical lines in the longitudinal profiles mark the locations of major water reservoirs along each river.**

The Gloria storm caused peak fluvial discharges across several watercourses in the region. Although these discharges were not extraordinary for most rivers, significant overflows occurred in the *Tordera* river.

Regarding the *Tordera*, an unregulated river, the peak flows in this river basin are recurring phenomena that occur periodically. Several episodes of significant magnitude occurred in the mid-20th century, while moderate-intensity peak flows were common in the late 20th and early 21st centuries (Farguell, 2020; Pavón and Panareda, 2020). The Gloria storm caused the third-largest rise recorded in historical data from the 1970s and 1980s, thought it was far on time from earlier rises. Peak flows in unregulated river basins are mainly associated with the storm precipitation patterns, however the intensification of river runoff during the Gloria storm episode was also due to prior soil moisture of the headwater area during the preceding months (Pavón and Panareda, 2020). The *Tordera* river had return periods of 10 to over 20 years (Fig. 8), with historical records explaining the shorter return periods, while the intensity of the rainfall combined with pre-storm conditions contributed to the higher ones.




### 4.2.3 Coastal hazard

The spatial distribution of the maximum significant wave height return period reached during Storm Gloria at 52 grid nodes along the Catalan coastline, as well as the duration of exceedance over return periods of 100, 200, 300, and 400 years during the storm event (from January 19th to 24th, 2020) is presented in Fig. 9.


Return periods and the exceedance times varied across the coastline, showing distinct trends in the northern (*Girona* province), central (*Barcelona* province), and southern (*Tarragona* province) areas (Fig. 9). The northern coastline exhibited the smallest return periods, between 100 and 300 years, with a maximum of 263 years. Exceedance times in this region were also brief, with 100-year return period exceedances lasting less than 5 hours and 200-year return period exceedances lasting

around 1 hour. In contrast, the central *Barcelona* coastline exhibited moderately higher return periods and longer exceedance durations. SIMAR estimations in this area surpassed the 100-year return period for extended durations exceeding 5 hours. Most locations also exceeded the 200-year return period for around 5 hours, while a few briefly experienced higher return periods for 1 to 2 hours, reaching a maximum of 488 years. Conversely, the southernmost region (*Tarragona*) displayed the highest return periods, reaching up to 824 years and surpassing 500 years at 5 SIMAR grid nodes, although with moderate

exceedance durations (lasting around 5 to 10 hours for 100-year return period and 1 to 5 hours for the higher return periods).

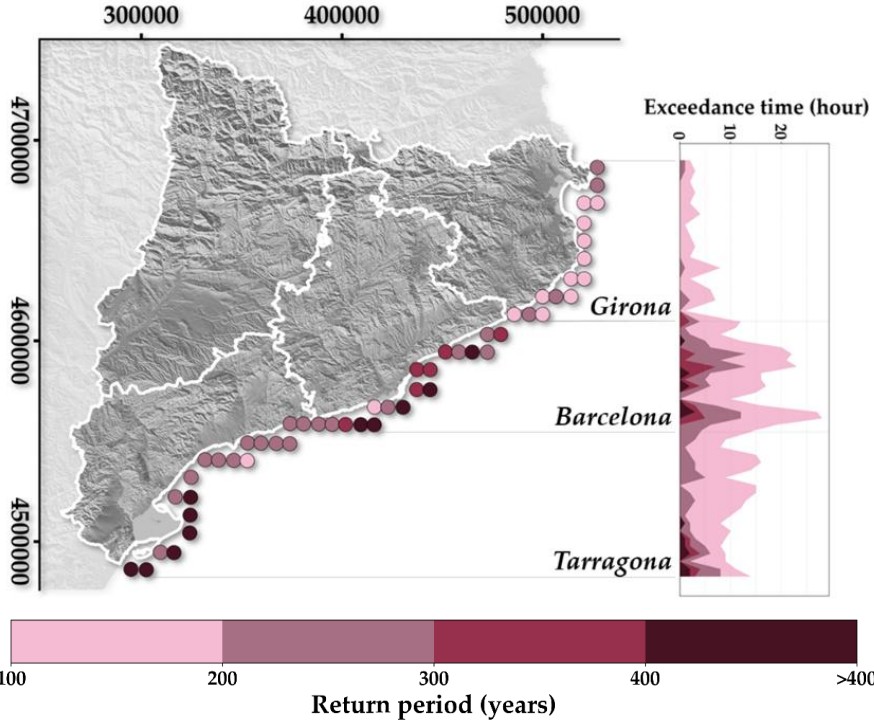

**Figure 9. Geographical distribution of the significant wave height return periods related to the coastal hazard analysed and the hours of exceedance for the SIMAR grid nodes along the three coastal provinces.**





Spatial analysis revealed a distinct north-south gradient in return periods of significant wave heights, suggesting potential geographical variations in historical records across these regions. In the north, data from the *Begur* buoy (see location in Fig. 1) showed that during the Gloria storm, significant wave heights remained above 4 meters, peaking at nearly 8 meters (Fig. 2c). Historical records suggest that such coastal conditions are common in the area, resulting in lower return periods of about 200 years, as stated by Jiménez Quintana (2020), and consistent with the findings of this paper. In contrast, along the southern coasts, the *Tarragona* buoy recorded significant wave heights remaining above 2 meters for three consecutive days during the Gloria storm, and reaching 4 meters for the first time since its installation in 1992. This indicates that such events are much rarer in the south compared to the north (Pintó et al., 2020; Puertos del Estado, 2024). Consistent with this paper's findings, Jiménez Quintana (2020) reported that the significant wave height return period for the southern coast exceeds 300 years. It is also worth noting that de Alfonso et al. (2021) and Canals and Miranda (2020) highlighted the exceptional duration of the coastal storm in their research.

## 4.3 Losses as function of annual occurrence

One goal of this study was to provide a general assessment of losses based on annual occurrence probabilities across multiple hazards, focusing on public loss compensation for typically uninsured assets. The previous results showed that fully public compensation was the only source of recovery funding for the "Agribusiness", "Environment", "Water", and "Culture and Recreation" assets, and the primary source for "Transport". Therefore, these affected elements are the sole focus of the assessment in this section. Density plots in Fig. 10 illustrates the relationship between the logarithmic cost of damages and the standardized return period across different asset categories affected by coastal, fluvial, and meteorological hazards. Each plot represents data from a specific number of municipalities (denoted by "n").

The findings suggested that there is no direct correlation between the standardized return periods of hazards and recorded losses. Notably, coastal hazards showed a consistent pattern, with municipalities recording similar losses across both shorter and longer return periods. A similar trend is observed for fluvial hazards. However, meteorological hazard displayed a distinct clustering pattern, where municipalities experienced comparable return periods and similar levels of recorded losses. This trend is particularly noticeable in the "Transport" sector, where coastal hazards led to compensations of up to EUR 10 million across both low and high return periods, while meteorological hazards resulted in smaller compensations of around EUR 1 million, mostly at shorter return periods. A similar pattern was also visible for the "Water" and "Culture and Recreation" assets. These distribution differences reflect the varying nature of the asset's categories. For instance, in the "Transport" asset, meteorological hazards typically impacted smaller urban areas, mainly damaging streets, whereas fluvial and coastal hazards affected a broader range of more expensive infrastructure, such as harbors, highways, railways, etc. Similarly, in the "Water" asset, meteorological hazards primarily damaged sewage networks, while coastal and fluvial hazards affected more critical infrastructure, such as desalination plants and water treatment facilities. Despite the differences in damage patterns, the results highlighted that many essential public assets were highly vulnerable to





meteorological hazards. Even though this hazard tended to cause lower-cost damages, municipalities still faced some level of destruction at shorter return periods.

Another observation was that, for similar recorded costs, the return periods required to reach those costs were highest for
coastal hazards, followed by fluvial hazards, and lowest for meteorological hazards. For example, damage to "Water" infrastructure valued at approximately EUR 100,000 occurred at a standardized return period of 0.25 for coastal hazards, 0.15 for fluvial hazards, and close to 0 for meteorological hazards. The findings emphasized Catalonia's significant exposure to coastal and fluvial hazards, suggesting that protective measures were already in place to address these risks. The importance of fluvial and coastal hazards was also evident in the "Environment" asset category, where loss compensations
were used to cover damage to riverbeds and sand deposits in coastal areas. Beach sand and riverbeds play a crucial role in Catalonia, not only offering natural protection but also contributing to the region's environmental richness and recreational appeal, which are integral to its identity.

Several difficulties and uncertainties arose during the task of linking losses with return periods. They can be divided into
three topics: hazard assessment, the loss database, and exposure and vulnerability aspects. These factors increased the complexity of assessing public loss compensation for typically uninsured assets, particularly in the context of extreme events such as Storm Gloria, and will be described below.

Firstly, in hazard assessment, uncertainties emerged from the reliance on probabilistic methods for calculating return periods,
which inherently involved some degree of uncertainty. Extreme events like Storm Gloria, which deviated from typical patterns, added complexity to these estimations because of its rarity and unpredictability. Additionally, while the correlation between hazard occurrence probability and financial losses proved to be insufficient, other factors characterizing the hazards differently may need to be considered for more accurate damage assessment. Moreover, the complex interactions between different types of hazards also led to compounding and cascading effects not captured by return period calculations. The
second source of uncertainty stemmed from the loss database used in the assessment, where the collected data significantly influenced the accuracy of the financial loss assessment. A key concern was consistency, ensuring uniform and coherent reporting, though methods varied across state and regional agencies in Spain and Catalonia. Finally, assessment difficulties also arose from exposure and vulnerability aspects. While hazard characteristics play a significant role in determining financial losses, exposure and vulnerability factors can have an even larger impact. The degree of exposure depends on the
distribution of assets within hazard-prone areas, while vulnerability refer to the susceptibility of those assets to damage.





**Figure 10. Density plots of the logarithmic cost of damage versus standardized return period across different hazard types ("Meteorological", "Fluvial" and "Coastal") and impacted "Affected element" ("Water", "Environment", "Culture and Recreation", "Agribusiness" and "Transport"). Each plot shows the number of municipalities analyzed (n). The colour scale reflects the number of municipalities, with darker blue indicating a higher concentration of municipalities and lighter blue representing fewer municipalities.**





The spatial heterogeneity of exposure and vulnerability across municipalities made it difficult to generalize the correlation of the financial impacts of the Gloria storm based on return period. Each municipality have distinct infrastructure characteristics, asset distributions, vulnerability levels, and disaster preparedness programs, all of which influence the financial impact of the natural hazard event. This variation has to be accounted for in order to accurately estimate financial losses across the region. While efforts were made to normalize exposure and vulnerability parameters to improve

comparability, the results lacked precision. The following factors were used to make financial losses more comparable across municipalities: population (absolute and density), gross domestic product and land use data (percentage, area or density). Further attempts to subdivide or aggregate asset categories for more detailed analysis also yielded limited insights.

In conclusion, fully public compensations differed significantly from those of PPPs. The findings indicated a distinct cost

distribution, typically covering moderate expenses, with a prioritization based primarily on hazard type, followed by the affected element. Additionally, these funds did not appear to align with return period calculations, even when factors like exposure and vulnerability were considered. Instead, public compensation seemed to be guided by a standardized formal government procedure, considering the affected element strategic importance, and the overall need for recovery. Other elements, such as available funds, damage grids, and administrative considerations, likely also played a role in shaping the

final distribution of recovery resources.

## 5 Conclusions and Discussion

This study highlights the critical role of the public sector in managing financial losses following multi-hazard events, using Storm Gloria as a case study. Data on public financial support for post-disaster recovery, often underrepresented in disaster analyses, is essential for understanding compensation for uninsured asset damages and refining future recovery strategies. To

address this, we systematically collected and classified public-sector compensation data allocated for repairing and rebuilding direct tangible losses caused by Storm Gloria, selecting five entities to characterize these financial impacts. The resulting loss database, built on established methodologies, allows for flexible disaggregation and aggregation of categories, ensuring both adaptability and standardization. A post-analysis of the selected attributes and categories confirms their adequacy in describing the economic impact of the hazard event, based on extensive document collection. However, our

study aligns with previous research, such as De Groeve et al. (2013), which emphasizes the importance of incorporating additional attributes, such as geographical and temporal hazard information, as well as characteristics of affected elements, beyond simply identifying the event itself to improve impact assessments.

Storm Gloria brought severe meteorological, coastal, and fluvial hazards to Catalonia, marked by its exceptional intensity,

prolonged duration, and multi-hazard nature. The storm caused widespread damage across the region, with documented repair and rebuilding costs amounting to EUR 264 million, underscoring the storm's significant financial burden. While



comprehensive, our database likely reflects only a portion of the total costs, as it accounts solely for actual verified investments. The OCCC (2020) estimated total damages in Catalonia to exceed EUR 500 million, incorporating preliminary and unverified sources, suggesting actual costs were significantly higher. Among global public disaster databases, EM-DAT
recorded total damages of just USD 315,000 for Spain (CRED, 2024), far below our estimates, while the DesInventar Sendai database has not updated Spain's data since 2010, limiting comparative analysis. A main limitation of the present study is the lack of prior research on post-storm public compensation data, highlighting both the novelty and the importance of our investigation. Therefore, collecting public-sector compensation data provides a crucial basis for future research and should be extended to other multi-hazard contexts for validation and refinement.


Fluvial and coastal hazards were the dominant triggers, accounting for over 80% of total damages, while meteorological and slope hazards contributed to the rest. Gall et al. (2009) noted the challenge of "hazard bias" in databases, where certain hazard types may be over- or underrepresented. In this study, it remains unclear whether the lower recorded losses from slope and meteorological hazards reflect genuine lower impacts, under-prioritization, or limitations in evaluation methods.
Prior research in Catalonia indicates that flash floods tend to cause substantial tangible losses and mortality (Gil-Guirado et al., 2019; Llasat et al., 2013), with fluvial components causing significantly higher losses than marine components (Romero-Martín et al., 2024). However, all municipalities experienced some level of destruction from meteorological and slope hazards, emphasizing the need for comprehensive documentation of all losses. Reporting biases may also extend to affected elements, as specific damaged categories may receive preferential attention ("accounting bias"). Concerning the affected
elements, "Infrastructure" sustained the highest losses (42%, primarily in "Transport"), followed by "Economic" sector (28%, especially "Trade") and "Social" sector (20%, particularly "Residential"). Additionally, differences in recovery cost distribution, such as those for fluvial hazards and environmental impacts, suggested variations in compensation calculation or reporting methods.

Rebuilding and reconstruction costs for Storm Gloria were evenly split between fully public and public-private partnership funding, reflecting a global balanced approach to financial responsibility. Regarding this aspect our main observations include: (1) financial responsibilities were allocated according to sector-specific recovery needs, with fully public funding prioritizing community assets, public welfare and critical infrastructures; (2) fully public compensation focused fist on hazard-dependent cost assessments and assets, whereas PPP funding targeted affected elements regardless of hazard type
and, (3) fully public compensation followed standardized formal government recovery procedures.

Several municipalities were identified as potential multi-hazard areas, where overlapping hazards coincided with a higher number of affected elements and greater incurred costs. Our database identifies municipalities where multiple hazards occurred simultaneously but does not establish the cumulative, cascading or compounding action of all contributing
processes. As Kappes et al. (2012) noted, a common limitation of most disaster databases is their single-hazard focus, often



neglecting hazard interactions. Despite this prevalent approach, multi-hazard events, which account for nearly 59% of global economic losses according to EM-DAT (Lee et al., 2024), are crucial to consider in disaster impact assessments. The multi-hazard map generated from Storm Gloria is the first of its kind in Catalonia at municipality level, covering four different hazard types. Limited research has been conducted on multi-hazard mapping in Catalonia. Romero-Martín et al. (2024)
applied an index-based framework to assess integrated risk along the NW Mediterranean coast by aggregating major hydro-meteorological and marine hazards. Their findings identified the highest compound risk in densely urbanized coastal zones, such as the Ebro Delta and Barcelona, where high exposure, significant hazards, and vulnerability intersect, aligning with our findings. Similarly, Antofie et al. (2025) mapped multi-hazard exposure at a pan-European scale, examining relationships between assets (population and residential built-up areas) and various natural hazards, including landslides,
coastal flooding, river flooding, earthquakes, wildfires, and subsidence. Their findings highlight Barcelona and northern Girona as multi-hazard hotspots for residential built-up areas, further supporting our results.

Analyzing public-sector compensation data revealed no clear correlation between hazard likelihood and recorded losses despite multiple analyses. Notably, coastal and fluvial hazards displayed consistent loss patterns across varying return
periods, while meteorological hazards exhibited distinct clustering. Despite generally lower-cost damages, meteorological hazards still caused significant destruction in shorter return periods, highlighting asset vulnerability. However, uncertainties and challenges arose in linking losses with return periods, particularly in hazard assessment, loss databases, and exposure and vulnerability aspects. These complexities complicate public loss compensation estimates for typically uninsured assets, particularly in extreme events like Storm Gloria. The lack of correlation between compensation and hazard levels further
underscores the unique approach of fully public funding in prioritizing recovery efforts. Similarly, Rivas et al. (2022) examined oceanographic parameters influencing storm damage along the eastern Cantabrian coast, finding no direct link but emphasizing that severe coastal damage results from a combination of critical conditions, a conclusion fully supported by our study.

This study provides significant insights into public-sector compensation for disaster recovery and shows that efforts must be done to adequately face the challenge of future changes. As Jiménez et al. (2012) noted, past analyses of marine storm damage along the Catalan coast reveal that, even without increased storm frequency, rising exposure levels and growing coastal vulnerability, exacerbated by narrowing beaches, have led to escalating damages. Additionally, increased urbanization in flood-prone areas, permitted by local administrations in recent decades, has significantly shaped current
flood risk scenarios along the Spanish Mediterranean coast and along Spanish rivers (Cánovas-García and Vargas Molina, 2025; López-Martínez et al., 2020). These factors highlight the need for improved disaster preparedness and recovery strategies to mitigate future risks effectively.



**Code and Data availability**

The code and the data can be provided by the authors upon reasonable request.

**Author contribution**

NPR, MH and NL conceptualized the project and its methodology. NPR performed the data curation, investigation, formal analysis, visualization, writing the first draft, review and editing. MH and NL completed the investigation, review and editing.

**Competing interests**

The authors declare that they have no conflict of interest.

**Acknowledgements**

This publication was developed within the EU PARATUS project. The author wishes to acknowledge and express sincere gratitude for the personal assistance provided by Vicente Medina, Marc Berenger and José Antonio Jimenez. The authors extend their gratitude to the UPC's Center of Applied Research in Hydrometeorology (CRAHI), the Meteorological Service of Catalonia, the Catalan Water Agency and the *Consorcio de Compensación de Seguros* for kindly providing data.

**Financial support**

This study has received funding from the European Union's Horizon Europe research and innovation programme PARATUS under grant agreement no. 101073954 — HORIZON-CL3-2021-DRS-01.

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
