# Peer review of "How are public compensation efforts implemented in multi-hazard events? Insights from the 2020 Gloria storm in Catalonia"

_EGUsphere, 2025_

## Author Response (AR1)

**Reply to reviewer comments on Egusphere-2025-1009**

We express our sincere thanks to the reviewer RC1 for his time, insights and constructive engagement with our manuscript. Below, we provide detailed responses to the reviewer comments. We believe that the suggested changes from the reviewer have improved the quality and argument of the manuscript and thank the reviewer again for his contribution.

**(RC1-00):** "The paper examines public-sector compensation in disaster recovery, focusing on the 2020 Gloria storm in Catalonia. It highlights the importance of public compensation for uninsured assets and provides insights into financial aid distribution for disaster recovery. The study reveals that fluvial and coastal hazards caused over 80% of recorded damages, with infrastructure sustaining the highest losses. Public funding prioritized community assets and critical infrastructure.

The paper details the data used and the methodology with great precision and presents clear and concise results. The case study seems very appropriate to me. In addition to good conclusions. For all these reasons, I recommend its publication. I simply add some personal recommendations that the authors may or may not consider:"

**(Reply to RC1-00):** We thank RC1 for the thorough and positive feedback. We are pleased to hear that the case study and methodology were well-received and that the conclusions were clear and valuable. We appreciate his recommendations and will carefully consider them in the revision process. Below we respond to each comment.

**(RC1-01):** "In line 226, Gumbel is mentioned but its use is not justified as it is done with GEV previously."

**(Reply to RC1-01):** Thank you for this comment. Indeed, the sentence in the original manuscript lacked sufficient detail. Based on the available data, different types of Generalized Extreme Value (GEV) distributions were used to calculate the return periods for each hazard. Since the IDF curves for rainfall data from Llabrés Brustenga (2020) were derived using a Gumbel distribution (Type I of the GEV family), we also employed a Gumbel distribution to calculate the return period of rainfall accumulation during the Gloria Storm event. We adapted the revised manuscript at the following points:

- We will amend the following sentence: "Since the IDF rainfall maps were derived using a Gumbel distribution fit (Llabrés Brustenga, 2020), we used the corresponding formula to calculate the return period of the rainfall accumulation".
- Additionally, to provide clarification, we will add the following sentence to the revised manuscript to clarify that the Gumbel is a type of GEV distribution: "This distribution includes the Gumbel (Type I) distribution, the Fréchet (Type II) distribution and the Weibull (Type III) distribution, respectively, when the shape parameter is equal to 0, greater than 0, and lower than 0".
- We will modify the sentence to clarify that different types of distributions will be fitted according to the data available: "Hence, depending on the data available, we apply different types of GEV distributions to characterize the Gloria storm, considering three main hazards: meteorological, fluvial, and coastal".

**(RC1-02):** "In figure 6, it would be advisable to add the letter labels to know what the caption refers to (a, b, c…)."

**(Reply to RC1-02):** We appreciate the reviewer's suggestion. As recommended, letter labels will be included to improve the clarity of Figure 3.

**(RC1-03):** "In the conclusions, add future contributions following the line of research."

**(Reply to RC1-03):** We agree that outlining future steps in relation to this research provides valuable context for the work completed until now. Therefore, we will include the following sentences in Section 5:

"In recent years, Machine Learning (ML) algorithms have emerged as a valuable tool for assessing the multiplicity of impacts that may affect a region. These methods effectively process large volumes of heterogenous data and model complex non-linear relationships among multiple factors. Due to the complexity of multi-risk systems, ML techniques are increasingly being used to investigate the connections between natural and human-driven pressures, helping to better understand the consequences of these interactions. Future research could build on this work by applying ML algorithms to the dataset developed here, aiming to uncover new insights into how public compensation mechanisms function in the context of multi-hazard events."

**(RC1-04):** "If possible, a graphical diagram of the methodology used in the paper."

**(Reply to RC1-04):** Thank you for the suggestion. We agree that including a figure would provide clearer guidance for readers. We will include in the manuscript the following Figure 3.

[Figure]

Figure 3. Graphical diagram of the methodology.

Additionally, we will add a brief subsection "3.3 Losses as function of hazard occurrence" to the "Methods and Data" section to provide further detail on the third component of the methodology. This will enhance the manuscript by incorporating the following paragraph:

"3.3 Losses as function of hazard occurrence

A general assessment of losses as a function of hazard occurrence probability was determined. Losses were calculated by aggregating recorded costs at the municipal level for each asset and hazard type ("Hazard identification" and "Affected element"). For each municipality, losses were then aligned with a standardized return period. Return periods were determined using the median return period of 48-hour accumulated rainfall within each municipality for meteorological hazard, the closest upstream gauge station for fluvial hazard, and the nearest SIMAR point for coastal hazard. The

standardization was performed by normalizing return periods relative to their maximum values. To visualize the relationship between losses and hazard frequency, we generated separate density plots for each hazard and asset type."

**References**

Llabrés Brustenga, A.: Intensity - duration - frequency of rainfall in catalunya: maximum expected precipitation and idf relationship at high temporal and spatial resolution, https://diposit.ub.edu/dspace/bitstream/2445/151942/1/ALB_PhD_THESIS.pdf, Universitat de Barcelona, 2020.

This paper presents an analysis of public-sector compensation following Storm Gloria in Catalonia, Spain. The manuscript consists of three analyses focusing on: (1) losses, (2) hazards, and (3) the intersection of losses and hazards. A direct economic loss database was derived by categorizing public funds that were allocated towards recovery. The hazard analysis focused on three hazards (meteorological, coastal, and fluvial) and consisted of evaluating the spatial extent of hazards and determining hazard return periods. The intersection of losses and hazards evaluated the relationship between losses and hazard return period. The results shows that fluvial and coastal hazards are the major driver of losses for Storm Gloria, and that infrastructure sustained the largest losses. The manuscript shows no relationship between losses and hazard return period.

This manuscript is organized, well written, and easy to read. The reviewer recommends revisions prior to acceptance in Natural Hazards and Earth System Sciences. The reasons for this recommendation, and suggestions for improvement of the manuscript, are outlined below.

Thank you for your helpful and detailed comments and suggestions. We greatly appreciate the time and effort you dedicated to reviewing our manuscript. Your insights have been useful in improving the quality of our work. We have carefully considered all your comments and revised the manuscript accordingly. Below, we provide a point-by-point response to each of your observations, outlining the changes made and clarifying specific aspects as needed.

*Major comments:*

**(RC2-00):** Line 148: The introduction to the methods section outlines a three-step methodology; however, the methods section only consists of two subsections (3.1 Direction Loss Analysis and 3.2 Hazard Analysis). Is there anything that can be said in the methods about the third step in the methodology? This doesn't need to be long. It would help readers know what to expect further in the manuscript. As it stands now, line 151 (the one-sentence description of the third step) isn't clear to me.

**(Reply to RC2-00):** You are right, a subsection addressing the third step in the methodology is necessary for clarity and to better guide the reader through the structure of the manuscript. Accordingly, we will add a brief subsection "3.3 Losses as function of hazard occurrence" to the "Methods and Data" section to provide further detail on the third component of the methodology. This will enhance the manuscript by incorporating the following paragraph:

"3.3 Losses as function of hazard occurrence

A general assessment of losses as a function of hazard occurrence probability was determined. Losses were calculated by aggregating recorded costs at the municipal level for each asset and hazard type ("Hazard identification" and "Affected element"). For each municipality, losses were then aligned with a standardized return period. Return periods were determined using the median return period of 48-hour accumulated rainfall within each municipality for meteorological hazard, the closest upstream gauge station for fluvial hazard, and the nearest SIMAR point for coastal hazard. The standardization was performed by normalizing return periods relative to their maximum values. To visualize the relationship between losses and hazard frequency, we generated separate density plots for each hazard and asset type."

**(RC2-01):** Line 229: Why is the return period for meteorological hazards of each pixel assigned to the nearest return period available from the Catalan meteorological service (2, 5, 10, … 500-yr)? That is, in the example given, why is the interpolated return period of 29-year event re-assigned a return period of 20-

years? This does not appear to be consistently done across hazards (see line 255 where the return period for fluvial hazards is estimated as 12-years and not re-assigned).

**(Reply to RC2-01):** Thank you for raising this point. The decision to assign meteorological hazard return periods to the nearest values provided by the Catalan Meteorological Service was made based on expert judgment and using their classification methods. We recognize that this differs from the approach used for fluvial hazards and we will clarify this distinction in the revised manuscript, as follows:

"This approach follows criteria defined with the Catalan Meteorological Service, based on expert judgment and their classification thresholds."

**(RC2-02):** Line 269: Can the authors provide a justification for using Block Maxima method as opposed to something such as peaks over threshold?

**(Reply to RC2-02):** Thank you for this comment. Yes, both the *Peaks Over Threshold* and *Block Maxima* methods are commonly used for modeling extremes and estimating return periods. Given the relatively long dataset spanning 65 years, the differences between the two approaches were found to be minimal, and both tend to produce very similar results. Since the primary focus of the paper is on the dataset and damage compensations, we have not provided an in-depth discussion of the return period calculation methods in the manuscript. Nevertheless, we will clarify this justification in the revised version to better inform readers about our methodological choices, as follows:

"Given this relatively long time series, the BM method is appropriate for capturing extreme events consistently over time."

**(RC2-03):** Fig. 6f: Am I viewing this figure correctly in that the largest possible number of effected elements in each municipality is only 9? I would assume there are more elements/assets in a municipality. It would help readers if the total number of entries in the compiled database is clearly presented throughout the manuscript (maybe a new column Table 1 with "number of elements" or shown somewhere in Fig. 4?).

**(Reply to RC2-03):** We thank the reviewer for this observation. You are correct that the numbers shown on the map (Fig. 6f) represent the count of distinct second-level asset categories affected in each municipality, not the number of times each asset type was damaged. For example, in the case of Barcelona municipality, nine different second-level asset types were affected, but the map does not reflect how many instances of each asset type were damaged.

We agree that showing the total number of affected elements would enhance clarity for readers and provide deeper insight into the region's exposure to multi hazard events. However, due to the nature of the data we collected, including this information would not be appropriate or reliable. Some entities provided highly detailed data, for example, multiple entries for transportation infrastructure, with each individual repair recorded separately. Others, however, provided more aggregated data, for instance, a single entry summarizing all residential damages within a municipality. These differences in reporting levels make it difficult to ensure consistency or comparability across asset types and municipalities.

We consider this an important point to address in the manuscript, and will therefore include the following sentence highlighting the disparities in data reporting:

"Notably, Fig. 6f shows the number of distinct asset types affected in each municipality, rather than the total count of damaged elements, due to disparities in reporting detail."

**(RC2-04):** Fig. 7: Are the return period maps shown in Fig. 7 all produced using the same linear distribution that is shown in Fig. 3 (e.g., the 48-hr duration)? If so, why are the 1-hr and 12-hr maps using the 48-hr duration return period estimate? If the maps are produced using their own duration return period estimates, why are these so drastically different? I would assume that there should be more similarity in return periods across for different durations.

**(Reply to RC2-04):** Thank you for the insightful comment. Indeed, Figure 3 shows one example of a linear distribution for a single pixel; however, individual linear distributions were applied to each pixel. We will clarify this point and improve the explanation in the revised manuscript to avoid any confusion. Therefore, we will include this sentence:

"It is important to highlight that the linear distribution was derived separately for each pixel."

**(RC2-05):** Line 488: The authors suggest that there are "potential geographical variations in historical records across these [north-south] regions". Is this not something that can be directly determined rather than using terms "suggesting" and "potential"? It appears the authors have hourly wave data at 52 nodes (Table 2) and could perform an extreme value analysis at each node to determine if there are indeed north-south variations in the historical records. This isn't something that needs to be shown per se, rather something that could be definitively stated. It's additionally not clear why the return periods of significant wave heights (I'm assuming this is referring to the observed significant wave heights for Gloria) are used to justify this statement. That is, significant wave height can vary along the coast for a single storm.

**(Reply to RC2-05):** Thank you for this helpful comment. You are absolutely right, based on the data we have, we can make a more definitive statement rather than suggesting a potential geographical variation. The intention of the paragraph was to place the return period values into context and to show how exceptional (or not) Storm Gloria was compared to historical records, including previous maximum values. However, we agree that even outside of storm events, these two geographical zones (north and south) show noticeable differences in wave conditions.

We will revise the sentence accordingly to reflect this more clearly:

"Spatial analysis revealed a distinct north-south gradient in return periods of significant wave heights, indicating clear geographical variations in historical wave records across these regions."

- - - - - - - - - - - - - - - - - - - - - - - - - - - - - - - - - -

*Minor comments:*

1. Line 12: Could this sentence be rephrased? I was expecting "losses" to be computed from "damages"; however, I did not see this in the manuscript.

We will modify the sentence to: "Finally, the relationship between the observed losses and the return period of the triggering hazards is evaluated."

2. Line 29: The authors use USD here, but EUR elsewhere (e.g., line 289). If possible, I'd suggest staying consistent across this manuscript.

We will amend the following sentence: "EM-DAT recorded total damages of just EUR 270,000 for Spain"

3. Fig. 1: I'd suggest the authors consider adding a small inset map showing where Catalonia is located in Spain / Western Europe. This is not a requirement, but something to consider.

An inset map will be added to Figure 1 to show Catalonia's location within Spain (see below).

[Figure]

4. Line 286: I'd suggest the authors consider removing "on one side" (e.g., "…we present the density distribution (Fig. 4) and the ___ (Fig. 5)".) This currently reads as if the two plots shown are on the same figure.

We will remove accordingly to avoid confusion.

5. Line 318: "first-level" not "firs-level".

We will correct the spelling mistake.

6. Fig. 6: Each subplot is missing a, b, c, etc.

Labels will be added to each subplot in Figure 6 as suggested.

7. Fig. 8: Is it possible to label the other rivers that the authors discuss in the text? Namely the *Tordera*, *Fluvià*, and *Besòs*. This would make it easier for the readers to identify which rivers the authors are discussing without having to refer to Fig. 1. Otherwise, this is a great figure.

We thank the reviewer's suggestion. Labeling the rivers directly in Fig. 8 would indeed make it easier for readers to follow the discussion. We considered adding the river names, but given that the map is already quite dense with information, we decided instead to include the labels in Fig. 1.

To improve clarity, the caption of Fig. 8 will be updated to include a more detailed description and a reference to Fig. 1 to help guide the reader more easily, as follows:

"Figure 8. Geographical Distribution of the water discharge return periods related to the fluvial hazard analysed and the hours of exceedance for the Llobregat (A-A') and Ter (B- B') Rivers. From north to south, the rivers analyzed are Muga, Fluvià, Ter, Tordera, Besòs, and Llobregat (see Fig. 1). Vertical lines in the longitudinal profiles mark the locations of major water reservoirs along each river."

8. Fig. 10 and line 504: Can the authors provide more information on what is meant by "standardized return period"?

We appreciate the reviewer's comment. We will add a sentence in the subsection "3.3 Losses as function of hazard occurrence" of the "Methods and Data" section to provide further detail on the standardized return period. Therefore, we will include the following sentence:

"The standardization was performed by normalizing return periods relative to their maximum values."

9. Line 561: "have" should be "has".

We will adjust accordingly.

10. Line 618: Check on the word "fist". Could this be removed, or should this be "first"?

The spelling mistake will be corrected.

**Reply to reviewer comments on Egusphere-2025-1009 (Samuele Segoni)**

Dear Authors,

I was at EGU2025 and I had the opportunity of reading a poster based on this publication. I also got the chance to have an interesting discussion with the first author. Since I liked the work described in the poster very much, I was also very curious to check the submitted manuscript.

For what my opinion is worth (I'm not an official reviewer), I highly recommend the publication of this work, as I found it very interesting, based on a sound methodology, and original.

I just put forward a couple of comments that you are free to address or discard.

Thank you very much for your kind comments. It was a pleasure discussing our work with the reviewer at the EGU2025, and we are truly grateful for his encouraging feedback on the manuscript. We are especially grateful for his positive remarks regarding the originality and soundness of our methodology. We will carefully take his suggestions into account as we revise the manuscript. Below, we provide detailed responses to the comments he kindly shared.

**(CC1-00):** "I don't know how frequent similar events in Catalonia are, but one thing that maybe could be stated more clearly is that it is safe to assume that this disaster didn't come on top of another precedent disaster from which the study area hadn't fully recovered yet. This is to avoid complex compound effects among repeated shocks that stack each other in a non-linear way, complicating any mathematics beyond the analysis."

**(Reply to CC1-00):** Thank you for the comment, we completely agree. In fact, there was a storm in December 2019, just a few weeks before Storm Gloria. While it was significant, it was not nearly as exceptional in terms of intensity, duration, or spatial extent compared to Gloria.

Regarding the potential for compounding effects, we fully acknowledge that the temporal proximity of the two events may have amplified the impacts. The reviewer correctly emphasizes that compound effects over time are one of the reasons why it is difficult to establish a linear relationship between hazard levels and resulting damages. In our study, these interactions were neglected. Nevertheless, during data collection and classification, we made a conscious effort to ensure that all reported damages and associated costs could be attributed solely and exclusively to Storm Gloria. However, we cannot be certain whether some of the damaged elements may had already been partially compromised or deteriorated prior to the storm.

Another important point is that, throughout the data compilation process, we found strong evidence that impact attribution tends to be assigned to individual hazards rather than to multi-hazard scenarios. Reporting on multi-hazard risk management is not a straightforward task, and at present, disaster risk management in Catalonia is not sufficiently detailed to assess the systemic effects of multi-hazard scenarios. We hope that this research will help drive change and promote greater attention to multi-hazard approaches in future disaster risk management strategies.

In response to the reviewer's comment, we will enrich the discussion in the revised manuscript by including the following sentences in the Section of "Discussion and conclusions":

- "The temporal variation of exposure and vulnerability is another key factor that influences this correlation; however, it could not be considered in detail in this study."
- "However, uncertainties and challenges arose in linking losses to return periods, particularly due to limitations in hazard assessment calculations and the temporal and spatial variability of exposure and vulnerability."
- "Reporting on multi-hazard risk management is not a straightforward task."
- "There is strong evidence indicating that impact attribution is more often assigned to single hazards rather than to multiple hazards."
- "This work aims to support a shift toward more integrated disaster risk management by encouraging greater consideration of multi-hazard approaches in future strategies."

**(CC1-01):** "While reading the paper I was very curious of finding some explanations or speculations about the spatial pattern of damages: why did some areas receive more direct damage than others? This issue is partially addressed in the manuscript, and I think you correctly pointed out that the impacts of some hazard are widespread, while others are clustered around the spots where the most severe phenomena occurred. Here I would suggest a rapid search for possible correlations with soil sealing (or soil consumption or imperviousness). Indeed, in recent research of my group (**DOI** 10.1088/1748-9326/ad5fa1), we discovered that such impacts do not occur at random places, are not driven only by the severity of the driving hazardous process (e.g. rainfall or discharge return time), but depend (a lot!) on how much each municipality built buildings and infrastructure, and, more importantly, where the urbanization occurred (specifically, to what extent high hazard and medium hazard areas were spared or aggressed by urbanization). I see that this is partially beyond the scopes of the work, but I think it is relevant for discussion and conclusion, as it could be useful information to better address future intervention by both the public and private sectors."

**(Reply to CC1-00):** We thank the reviewer for this insightful comment. We fully agree that the severity of impacts is strongly influenced by the degree of urbanization of a municipality. As the reviewer rightly noted, soil sealing is a critical factor that can amplify the effects of extreme weather events. Although soil sealing analysis was beyond the main scope of this study, our findings indicate that more sealed areas often correspond to larger urban centers, which typically have greater administrative capacity to apply for public recovery funds. In contrast, smaller municipalities may lack skilled personnel to manage the complex and time-consuming application processes typically required for such funding. We agree that these aspects deserve further attention, and we will integrate this important perspective into the discussion and conclusions of the revised manuscript.

We will add the following sentence: "Geographically, damage appears concentrated in major cities, likely due not only to greater exposure and vulnerability (Gatto et al., 2024) but also to their stronger administrative capacity to apply for and manage public recovery funds."

*References*:

Gatto, A., Martellozzo, F., Ciulla, L., and Segoni, S.: The downward spiral entangling soil sealing and hydrogeological disasters, *Environ. Res. Lett.*, 19, 084023, https://doi.org/10.1088/1748-9326/ad5fa1, 2024.